# Physics of AMOC multistable regime shifts due to freshwater biases in an EMIC

Amber A. Boot[1] and Henk A. Dijkstra[1,2]

[1]Institute for Marine and Atmospheric research Utrecht, Department of Physics,Utrecht University, Utrecht, the Netherlands
[2]Center for Complex Systems Studies, Utrecht University, Utrecht, the Netherlands

**Correspondence:** Amber A. Boot (a.a.boot@uu.nl)

**Abstract.** The Atlantic Meridional Overturning Circulation (AMOC), an important circulation system that modulates the global climate, has been identified as a potential tipping element. To assess AMOC tipping, climate models are used that are known to have many biases and it is unknown how these biases affect AMOC stability. We focus here on freshwater biases over the Indian and Atlantic Ocean, as identified in CMIP6 models. Next, we use CLIMBER-X, an Earth System Model of intermediate complexity, to study the effect of biases in surface freshwater flux on AMOC tipping behavior. We introduce biases in the Indian and Atlantic Ocean and perform hysteresis experiments where we slowly ramp up the surface freshwater forcing in the North Atlantic until the AMOC collapses; subsequently, the forcing is reversed until the AMOC recovers again. We find that negative (positive) biases in the Indian Ocean make the AMOC more unstable (stable), whereas negative (positive) biases in the Atlantic Ocean make the AMOC more stable (unstable). When biases are introduced in both the Atlantic and Indian Ocean, the tipping point associated with the AMOC collapse is hardly affected. These results show that if the freshwater bias we applied in the Indian Ocean is larger than the one applied in the Atlantic Ocean, the AMOC is more stable in CLIMBER-X. For more reliable assessments of AMOC tipping under future emission scenarios, (freshwater) bias reduction in climate models is therefore thought to be essential.

## 1 Introduction

The Atlantic Meridional Overturning Circulation (AMOC) has been identified as a potential tipping element due to the possible existence of multiple stable equilibria (Lenton et al., 2008; Armstrong-McKay et al., 2022). Stommel (1961) already suggested the existence of multiple stable equilibria in a simple box model. Since then evidence of multiple stable equilibria have been found in models over the full complexity range (Rahmstorf, 1996; Rahmstorf et al., 2005; Dijkstra, 2007; van Westen and Dijkstra, 2023) as well as indications in paleoclimatic proxies (Broecker et al., 1985; Lynch-Stieglitz, 2017; Weijer et al., 2019). Currently, the AMOC is in a strong, so-called 'on' state, but studies suggest that it can also be in a weak or even collapsed 'off' state (Weijer et al., 2019; van Westen and Dijkstra, 2023). Tipping of the AMOC would have dire consequences for the climate system, ecosystems and society. It would lead to large scale cooling in the Northern Hemisphere, while the Southern Hemisphere warms (Jackson et al., 2015; Liu et al., 2017; van Westen et al., 2024), precipitation patterns shift (Stouffer et al., 2006; Vellinga and Wood, 2008; Liu et al., 2017; Orihuela-Pinto et al., 2022; van Westen et al., 2024) and wind patterns change

(Orihuela-Pinto et al., 2022). An AMOC weakening or collapse would also lead to local changes in sea level with potential increases in the Atlantic basin (Levermann et al., 2005; Yin et al., 2010; van Westen et al., 2024). Furthermore, the global carbon cycle (Zickfeld et al., 2008; Boot et al., 2024) and marine ecosystems (Schmittner, 2005; Boot et al., 2023, 2025) are (negatively) affected as well. Another threat is so-called tipping cascades where tipping of the AMOC might lead to tipping of other tipping elements such as the Amazon Rainforest and the West Antarctic Ice Sheet (Dekker et al., 2018; Sinet et al., 2023; Wunderling et al., 2024).

Measurements of the AMOC are currently too short to accurately say whether the AMOC is decreasing or not (Lobelle et al., 2020). However, there are studies that have looked at proxies of the AMOC strength. Several of these studies state that the AMOC has been declining in strength over the last 100 to 1000 years (Dima and Lohmann, 2010; Rahmstorf et al., 2015; Caesar et al., 2018, 2021), though there are also papers find no decline in the AMOC strength. (Worthington et al., 2021; Latif et al., 2022; Rossby et al., 2022; Terhaar et al., 2025). Under climate change, the AMOC is projected to decrease in CMIP6 models, but no full collapse is found before 2100 (Weijer et al., 2020). When simulations are extended past 2100, models can show an AMOC collapse (Romanou et al., 2023). Partly based on the CMIP6 models, the IPCC AR6 report (Intergovernmental Panel on Climate Change (IPCC), 2023) states that it is unlikely that the AMOC will collapse this century.

However, these models might not be suitable to make a good assessment about AMOC stability. They, for example, struggle to represent past AMOC changes accurately (McCarthy and Caesar, 2023), and often do not include Greenland Ice Sheet melt. Recent studies contradict the AR6 report and suggest that the probabilities of a collapse are much higher than previously thought (Michel et al., 2022; Ditlevsen and Ditlevsen, 2023; van Westen et al., 2024). One reason of the underestimation of collapse probability in previous studies could be that biases in CMIP6 models lead to a too stable AMOC (Liu et al., 2017). An important metric in this regard is the freshwater transport by the AMOC over 34°S indicated here by $F_{ov,S}$, which is an indicator of the sign and strength of the salt-advection feedback (Vanderborght et al., 2024). It was earlier suggested that $F_{ov,S}$ is a potential indicator whether the AMOC is in a monostable regime ($F_{ov,S} > 0$) or in a multistable regime ($F_{ov,S} < 0$) (Dijkstra, 2007; Weijer et al., 2019). However, recent results showed that even AMOC states with $F_{ov,S} > 0$ can be in a multistable regime van Westen and Dijkstra (2023). From observations, $F_{ov,S}$ is negative (Bryden et al., 2011; Garzoli et al., 2013; Arumí-Planas et al., 2024), meaning that the salt-advection feedback is destabilizing the AMOC. In most CMIP3 (Drijfhout et al., 2011), CMIP5 (Mecking et al., 2017), and CMIP6 (van Westen and Dijkstra, 2024) models, $F_{ov,S}$ is positive, which suggests that the salt-advection feedback is stabilizing the AMOC. In van Westen and Dijkstra (2024), the biases in $F_{ov,S}$ were attributed to biases in the Indian Ocean which potentially make the AMOC more stable (Dijkstra and van Westen, 2024). There is, however, some criticism on using $F_{ov,S}$ as a stability indicator, since not all models show a clear relation between AMOC variations and $F_{ov,S}$ sign (Haines et al., 2022; Jackson et al., 2023a). Also the effect of salinity biases in the North Atlantic and its relation to deep convection on the AMOC have been studied (Danabasoglu et al., 2014; Heuzé, 2021; Jackson and Petit, 2022; Jackson et al., 2023b), which it is found that especially the salinity in the Labrador Sea influences important AMOC characteristics. Another important bias is the double ITCZ bias that is present in most CMIP models (Tian and Dong, 2020), and this bias has already been suggested to be a reason for a too stable AMOC (Liu et al., 2014).

In this study, we thoroughly investigate the effect of freshwater biases in the Indian and Atlantic Ocean, representing the double ITCZ bias, on AMOC stability in the intermediate complexity Earth System Model (EMIC) CLIMBER-X (Willeit et al., 2022). We perform a large set of hysteresis experiments with model configurations where artificial positive and negative freshwater biases have been introduced to assess the effect of these biases on the multiple equilibria regime of the AMOC.

## 2 Methods

### 2.1 CLIMBER-X

CLIMBER-X consists of components simulating the atmosphere (SESAM), land (PALADYN), sea ice (SISIM), and ocean (GOLDSTEIN). There are also components available within CLIMBER-X for ocean biogeochemistry (HAMOCC) and ice sheets (SICOPOLIS or Yelmo) but these are not used in this study. All submodules are run on a rectilinear 5° by 5° latitude-longitude grid. Due to this low resolution, we can simulate almost 10,000 model years per day which allows us to do many experiments to systematically study the AMOC multistable regime. Below a short description of the atmosphere, ocean, sea-ice and land models is provided, but for a more thorough description of the model we refer the reader to Willeit et al. (2022).

The atmosphere model in CLIMBER-X is the Semi-Empirical dynamical Statistical Atmosphere Model (SESAM). During the development of SESAM, extensive observational data were used as well as results from Global Climate Models (GCMs). SESAM can be classified as a 2.5D model where all prognostic variables (e.g. temperature, specific humidity, and eddy kinetic energy) are determined on a 2D grid and where the vertical dimension is purely diagnostic. The 3D structure of relative humidity and temperature are estimated using assumptions about the general vertical structure in the atmosphere of these variables, and the 3D structure of the wind is approximated using the thermal wind balance. Certain diagnostic variables, i.e. water transport, horizontal energy transport and vertical fluxes of longwave radiation, are determined using this 3D structure. Longwave radiation fluxes take several greenhouse gases into account, i.e. $CH_4$, $N_2O$, CFCs, $O_3$ and $CO_2$, as well as dust particles and sulphate aerosols. Of these, the $O_3$ and sulphate aerosol fields need to be prescribed to the model. Clouds are also represented in SESAM with one cloud layer having variables such as cloud fraction, cloud top height and cloud optical thickness.

The ocean model in CLIMBER-X is based on the GOLDSTEIN model (Edwards et al., 1998; Edwards and Shepherd, 2002; Edwards and Marsh, 2005). A major change compared to the original GOLDSTEIN model is that for the use in CLIMBER-X the equations are dimensionalized. GOLDSTEIN is run with 23 non-equidistant vertical layers. Horizontal velocities are determined using a frictional-geostrophic balance, the continuity equation is used to diagnose vertical velocities, and hydrostatic balance is assumed. The model uses a rigid-lid approximation, meaning that freshwater fluxes at the surface are transformed into virtual salt fluxes. For each time step, the virtual salt flux is corrected such that the globally integrated flux is equal to zero to conserve salinity. A major drawback of frictional-geostrophic balance-based models is that the Antarctic Circumpolar Current is too weak due to too strong momentum damping (Edwards and Marsh, 2005; Müller et al., 2006).

The Simple Sea Ice Model (SISIM) is the sea-ice model employed in CLIMBER-X. It represents one snow layer on top of one ice layer. The snow layer can accumulate and melt, and if the layer gets deeper than 1 m, the excess becomes ice. As

described above, the sea ice can accumulate due to a deep snow layer, and can experience melting from above and below. The sea-ice layer can also increase due to accretion from below. The freezing temperature of the seawater is dependent on the local ocean salinity through a non-linear relation. The sea ice is also allowed to drift, and the corresponding velocities are determined using an elastic-viscous-plastic rheology (Hunke and Dukowicz, 1997; Bouillon et al., 2009). SISIM also acts as a coupler between the atmosphere and ocean, including sea-ice free regions of the ocean.

PALADYN (Willeit and Ganopolski, 2016) is the land model of CLIMBER-X and models fluxes of energy and water between the atmosphere, the land surface and the soil. The terrestrial carbon cycle is represented and includes dynamical vegetation. In total there are five different vegetation types: needleleaf trees, broadleaf trees, shrubs, C3-type grass and C4-type grass. Besides these vegetation types, the land surface can be classified as bare soil, land ice and lakes. In the soil, temperature, water and carbon are solved for in five vertical layers, and permafrost is explicitly represented.

## 2.2 Experimental set up

In all simulations presented below, we apply a freshwater forcing between 20°N and 50°N in the Atlantic Ocean (see grey region in Fig. 1b) with a strength $F_H$. To conserve salinity, the freshwater forcing is compensated for by removing freshwater from the surface ocean globally. We first increase the freshwater forcing at a rate of 0.05 Sv/kyr until the AMOC collapses. From a collapsed state, we linearly decrease the forcing again at the same rate until the AMOC recovers. The resulting hysteresis diagram for our baseline case, for the standard values of the parameters in CLIMBER-X, is shown in Fig. 1a. For $F_H = 0$, the AMOC strength for this case is about 20 Sv on the forward branch (drawn curve) and with increasing $F_H$ it collapses near the point $S_1$. The simulation of the backward branch (dashed) shows an AMOC recovery near the point $S_2$. There are two main reasons why the hosing location was chosen: (1) this region is used in many other studies using a similar experimental setup (Rahmstorf et al., 2005; Hawkins et al., 2011; van Westen and Dijkstra, 2023), and (2) by chosing this region we do not directly hose the deep convection sites allowing internal feedbacks to be more dominant in the case of an AMOC collapse compared to the forcing. The hosing rate is chosen such that it is slow enough that the model does not deviate too much from its equilibrium, while fast enough to still be computationally feasible.

To motivate the applied biases, we consider the freshwater flux (P - E) biases in models (case BASE in CLIMBER-X and CMIP6) with respect to observations. For the latter, we take the observation based HOAPS4.0 dataset (Andersson et al., 2017) over the period from 1994 to 2014 (21 years). The full list of 32 different CMIP6 models of which we consider the historical simulations is shown in Table A1. For the CMIP6 models we use the precipitation ('pr') and evaporation ('evspsbl') variables to determine P - E, and for the HOAPS dataset we use the E - P ('EMP') data. We average both the HOAPS dataset and model data over the pink and yellow regions indicated in Fig. 1b. Most models (27 out of 32; Fig. 3) have a positive bias in the Indian Ocean, meaning that the net effect of this bias is to freshen the Indian Ocean. Most models have a bias of approximately 0.5 mm/day in the Indian Ocean, but the spread is quite large due to a few outliers. 20 out of 32 models have a negative bias in the Atlantic Ocean. The spread for the Atlantic Ocean biases is smaller compared to the Indian Ocean biases. However, there is more variation in the exact bias strength compared to the Indian Ocean where most models align around 0.5 mm/day.

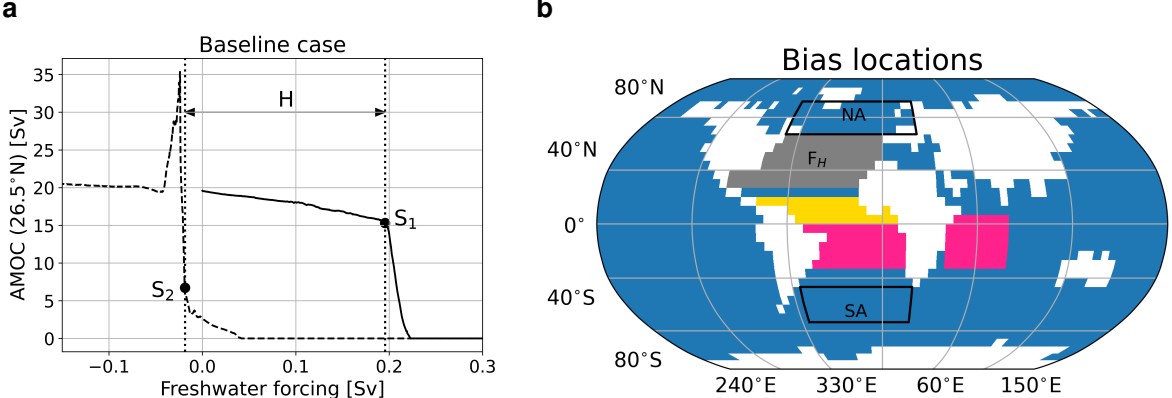

**Figure 1.** (a) Hysteresis diagram of the baseline case (BASE) with freshwater forcing $F_H$ in Sv on the x-axis and AMOC strength at 26.5°N in Sv on the y-axis. The solid line represents the forward branch, and the dashed lins the backward branch. $S_1$ and $S_2$ are the collapse and recovery tipping points and H is the hysteresis width as defined in Section 2.3. (b) Locations where the biases are deployed are denoted by pink and yellow. Biases in the yellow region are of opposite sign as in the pink region and 2/3 of the amplitude. Two boxes used for later analysis in Section 3.3, where variables are averaged over a North Atlantic (NA) and South Atlantic (SA) box, are also shown. The hosed region is in grey ($F_H$).

Motivated by these CMIP6 results (Fig. 2), we add positive and negative freshwater biases in the Indian and Atlantic Ocean in CLIMBER-X. Three different set of simulations are performed where (I) biases are introduced in the Indian Ocean, (A) biases are introduced in the Atlantic Ocean, and (IA) biases are introduced in both the Indian and Atlantic Ocean. Biases are introduced with 6 different strengths: +0.75 mm/year, +1.50 mm/year and +3.00 mm/year for the positive biases, and the negative equivalent of those for negative biases. This means that we performed in total 19 different simulations. In the Indian Ocean, the biases are introduced between 5°N to 25°S and 40°E to 80°E (Fig. 1b). In the Atlantic Ocean the biases are introduced in two sections. The southern section in the Atlantic (0°N to 25°S; pink section in Fig. 1b) receives the bias strength as mentioned above. The northern section (0°N to 15°N; yellow section in Fig. 1b) receives two third of the bias strength and of opposite sign to represent the double ITCZ bias found in many CMIP models (Tian and Dong, 2020). Before the hysteresis experiments are performed, a 10,000 year simulation is run to get the model in a new equilibrium after introducing the biases. In the text we will refer to the different simulations by their bias location and bias strength. For example, I(+3.00) represents the set up with a positive bias of 3.00 mm/day in the Indian Ocean, and IA(-0.75) represents the set up with a negative bias of 0.75 mm/day in both the Indian and Atlantic Ocean. For each of the 19 cases, we performed the same hysteresis experiment as for the baseline case, and the model is set up such that all cases have the same total salt content in the ocean at all times.

### 2.3 Tipping point detection

As can be seen in Fig. 1a, $S_1$ is the location of the tipping point representing the transition from the AMOC on to the AMOC off state, and $S_2$ is the location of the tipping point representing the transition from the off to the on state (Fig. 1a). We define

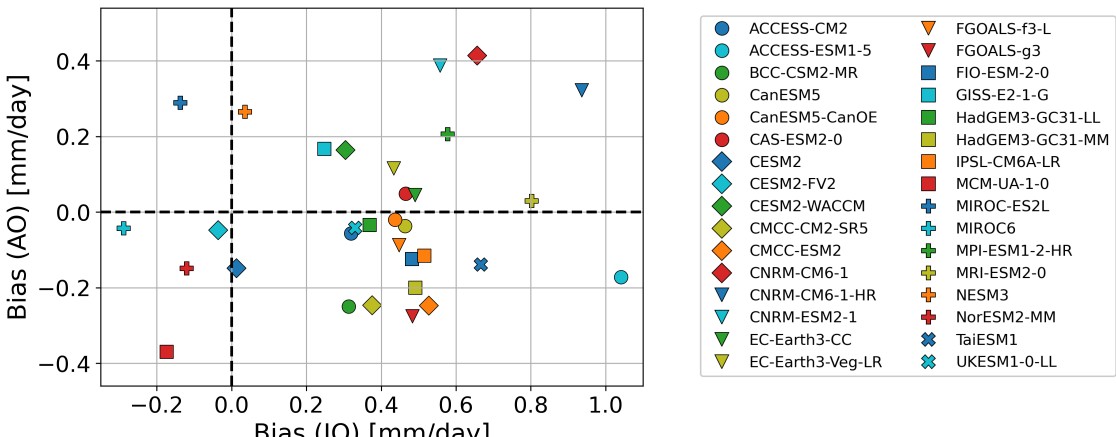

**Figure 2.** Biases in the Indian Ocean (x-axis) versus biases in the Atlantic Ocean (y-axis) in mm/day for 32 CMIP6 models. Atlantic Ocean biases are integrated over both the pink and yellow regions in Fig. 1.

the hysteresis width $H$ (in Sv) as the width in freshwater forcing $F_H$ between the tipping points representing the transition from the on to the off state and vice versa, i.e.

$$H = S_1 - S_2. \tag{1}$$

145  We will determine the shifts of H, $S_1$ and $S_2$ in the different experiments relative to REF, e.g. $\Delta H_x = H_x - H_{REF}$, where x represents an experiment with a freshwater forcing bias.

To determine the precise location of the tipping points $S_1$ and $S_2$, we employ a method based on detecting change points using the pruned exact linear time (PELT) method (Killick et al., 2012). In this method we are able to set the minimum time between two change points which allows us to tune this method to some extent. In our time series, as the AMOC is collapsing

150  or recovering, the time between change points decreases and converges towards the chosen minimum time. The minimum time between two change points for an AMOC collapse is set to 20 years, and the minimum time for an AMOC recovery is set to 5 years. We define a threshold for time between change points, i.e. 120 years for an AMOC collapse and 20 years for an AMOC recovery. Note that these threshold values are used for tuning of the method. When the time between change points becomes lower than such a threshold ($t_{th}$), we define the location of the tipping point as

155  $$s = X(t_{th}) - 0.5\Delta X, \tag{2}$$

where s is the tipping point in time space, X is the time of the change point at $t_{th}$, and $\Delta X$ is the time between the change point at $t_{th}$ and the next change point. Note that this method is subject to some subjectivity through the tuning. Hence, all tipping points are also checked visually to see whether the methodology gives reasonable results. For the IA(+3.00) simulation this method fails and the tipping points are determined visually.

## 3 Results

### 3.1 AMOC - $F_{ov,S}$ relation

The applied biases change the AMOC as well as $F_{ov,S}$ of the equilibrium states for $F_H = 0$. An overview of the equilibrium AMOC strength at 26.5°N and $F_{ov,S}$ of the different cases is presented in Fig. 3. Case BASE is within observational bounds of $F_{ov,S}$ but simulates a slightly too strong AMOC compared to observations. Negative biases in the Indian Ocean (cyan markers) decrease the AMOC strength and lead to a more negative $F_{ov,S}$, while positive biases (blue markers) lead to a stronger AMOC with a more positive $F_{ov,S}$. Biases in the Atlantic Ocean have an opposite effect. Negative biases (orange markers) increase the AMOC strength and lead to larger $F_{ov,S}$ and positive biases (red markers) lead to a weaker AMOC and more negative $F_{ov,S}$. A(+3.00) shows a break in the trend since $F_{ov,S}$ is larger than in A(+1.50). For the combined biases a clear non-linear relation is seen. Negative biases (olive markers) show a stronger AMOC and more positive $F_{ov,S}$ for stronger biases, whereas positive biases (green markers) show a weaker AMOC and more positive $F_{ov,S}$ for stronger biases. This relation is caused by the competing effects of the biases in the Indian Ocean and Atlantic Ocean on the AMOC strength and $F_{ov,S}$.

We can explain this behavior in $F_{ov,S}$ as follows: for the Indian Ocean, if a positive bias is applied, the Indian Ocean becomes more fresh, which is transported towards the Atlantic section of the Southern Ocean. From here, less saline water is advected into the Atlantic basin across 35°S, effectively increasing the freshwater transport and therefore $F_{ov,S}$. The AMOC strengthens for positive biases because the density in the South Atlantic decreases due to freshening which increases the meridional density difference between the North and South Atlantic. When positive biases are applied in the Atlantic Ocean, the Atlantic basin becomes fresher, which reduces the AMOC strength and consequently also the freshwater transported by the overturning circulation.

### 3.2 AMOC multistable regime

From Fig. 1a, we see that for case BASE the AMOC tips at a freshwater forcing of 0.1956 Sv ($S_1$), and recovers at a freshwater forcing of -0.0187 Sv ($S_2$) resulting in a hysteresis width H = 0.2142 Sv. CLIMBER-X simulates a full AMOC collapse with no overturning in the North Atlantic in the off-state (Fig. A1).

For the cases with biases in the Indian Ocean the hysteresis diagram shows a shift with respect to case BASE (Fig. 4a). Positive biases (i.e. a too fresh Indian Ocean) cause the hysteresis diagram, i.e. both $S_1$ and $S_2$, to shift towards larger freshwater forcing, whereas for negative biases, the hysteresis diagram shifts towards smaller forcing. The biases in the Atlantic Ocean mainly cause a shift of $S_1$ where negative biases cause a shift towards larger forcing and positive biases towards smaller forcing. $S_2$ does not shift by a lot in most experiments except for A(-3.00). This means that the hysteresis width increases (decreases) under negative (positive) biases. Both A(+1.50) and A(+3.00) show a two-step tipping. At the markers of the tipping points (around 0.08 Sv), deep convection in the subpolar gyre collapses. This represents a weak AMOC state as found earlier in CLIMBER-X (Willeit and Ganapolski, 2024). The AMOC weakens to about 8 Sv and shows relatively strong internal variability. Around a freshwater forcing of 0.2 Sv the AMOC fully collapses since at this point deep water formation in the Greenland-Iceland-Norwegian (GIN) Seas ceases, implying that deep water formation in the GIN Seas is vital in CLIMBER-X

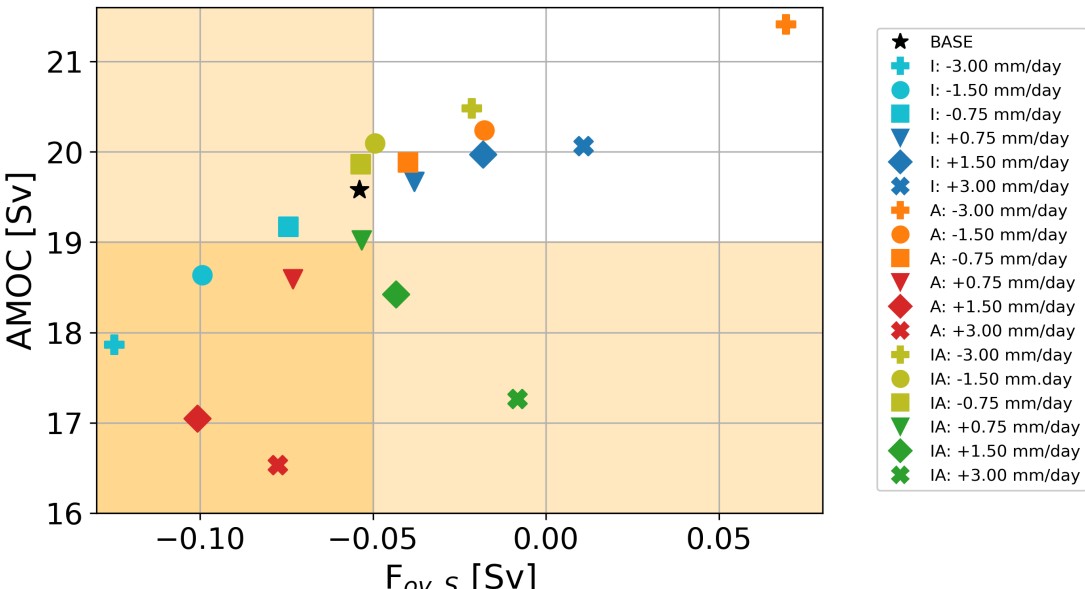

**Figure 3.** AMOC – $F_{ov,S}$ relation in equilibrium for the different simulations with the AMOC strength at 26.5°N in Sv on the y-axis and $F_{ov,S}$ in Sv on the x-axis. Yellow shading in the background represents observational bounds for the AMOC strength (Smeed et al., 2018; Worthington et al., 2021) and $F_{ov,S}$ (Garzoli et al., 2013; Mecking et al., 2017; Arumí-Planas et al., 2024); due to the grid in CLIMBER-X, $F_{ov,S}$ is determined at 35°S.

to sustain an AMOC. For the combined biases (IA), there is mostly a shift of $S_2$ whereas there is hardly a shift in $S_1$ (Fig. 4c). Negative (positive) biases cause an increase (decrease) of the multistable regime by moving $S_2$ towards more negative
(positive) values. Just as A(+1.50) and A(+3.00), IA(+1.50) shows a two-step tipping.

The change in location of the tipping points and the change in the hysteresis width is summarized in Fig. 5. For the Indian Ocean biases, $\Delta H$ is close to 0, with small negative values for negative biases, and small positive values for positive biases. For the Atlantic Ocean biases we can see that most markers fall along the $S_2 = 0$ contour line. However, for larger negative biases (orange markers) the deviation from this line increases. For the simulations with biases in both basins we see that the green
and olive markers are close to the line $S_1 = 0$. There is a small negative shift for negative biases and a small positive shift for positive biases. This means that with respect to the collapse tipping points, the Indian and Atlantic Ocean biases compensate each other about linearly.

### 3.3 Analysis

To explain the behavior seen in Section 3.2, we will look at surface properties in a North Atlantic box (50°N - 70°N; 70°W -
25°E) and a South Atlantic box (35°S - 55°S; 55°W - 20°E). We do this because these are the regions where we have isopycnal outcropping relevant for the AMOC strength (Wolfe and Cessi, 2015), and in CLIMBER-X the AMOC is strongly related to

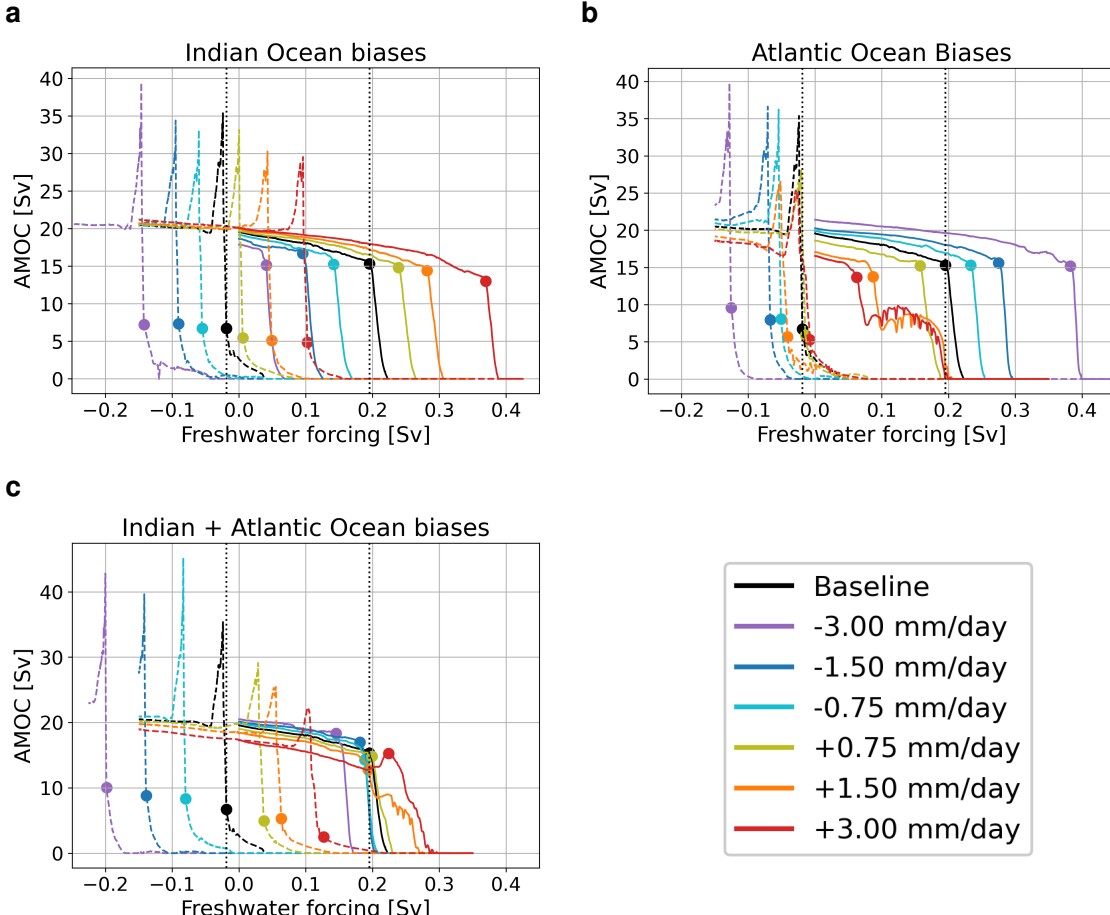

**Figure 4.** Hysteresis diagrams for the different simulations with AMOC strength at 26.5°N in Sv on the y-axis and the strength of the freshwater forcing in Sv on the x-axis. The black line represents the baseline case. Solid lines represent the forward simulation, i.e. under increasing forcing, and dashed lines represent the backward simulation under decreasing forcing. Markers represent the location of the tipping points determined by the method described in Section 2.3. The vertical black dotted lines represent the location of the tipping points in BASE. (a) Results for the Indian Ocean biases. (b) Results for the Atlantic Ocean biases. (c) Results for the Indian and Atlantic Ocean biases.

the meridional density gradient over the Atlantic (Fig. 6). Specifically we look at Sea Surface Temperature (SST), Sea Surface Salinity (SSS) and surface density for the equilibrium states at $F_H = 0$ (Fig. 7). Additional results, showing these variables along the hysteresis diagrams can be found in Fig. A2 to Fig. A4.

Negative biases in the Indian Ocean make the Indian Ocean more saline. This increase in salinity is transported towards the South Atlantic (cyan markers Fig. 7b). Here it increases the density which decreases the meridional density gradient between

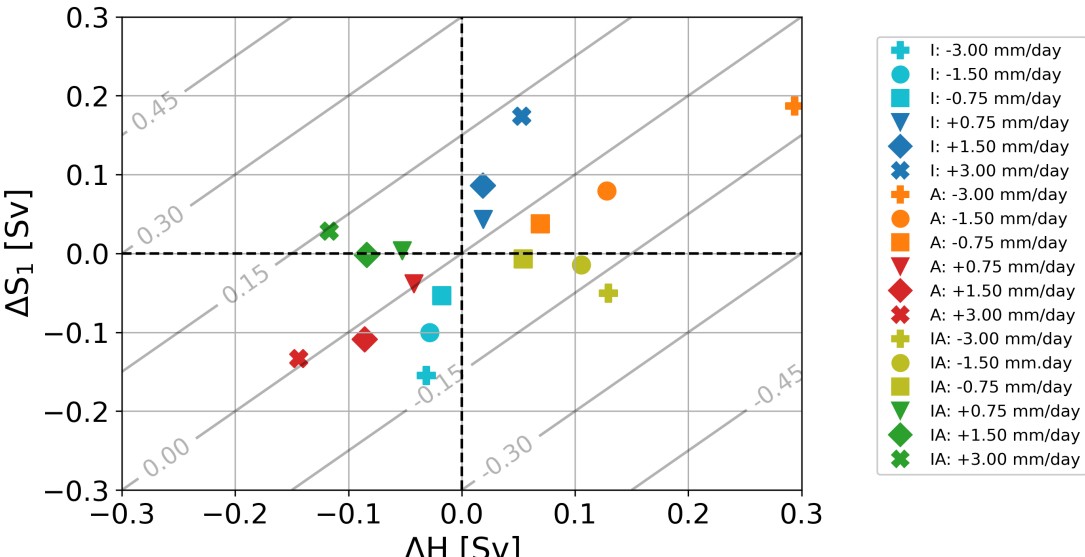

**Figure 5.** Overview of the change in hysteresis width ($\Delta H$) and change in location of the collapse tipping point ($\Delta S_1$) in freshwater forcing space in Sv. Black dashed lines represent $\Delta H = 0$ (vertical) or $\Delta S_1 = 0$ (horizontal). Contour lines in the background represent the shift of the recovery tipping point ($\Delta S_2$) in Sv.

the North Atlantic and South Atlantic (Fig. A3g). Due to this decrease, the AMOC weakens which reduces the transport of heat and salt towards the North Atlantic causing cooling and freshening (cyan markers Fig. 7a). For the density the freshening signal is dominant meaning that density decreases in the North Atlantic (cyan markers Fig. 7a). Because of the lower surface

density, a smaller value of $F_H$ is required to block the isopycnals from outcropping in this region. This explains the shift of $S_1$ towards smaller values of $F_H$ (Fig. 4a). For positive biases, the Indian Ocean freshens which results in the opposite response to the one described above (blue markers Fig. 7).

Negative biases in the Atlantic Ocean make the Atlantic basin more saline including an increase in salinity in the North Atlantic (orange markers Fig. 7a). This increases the meridional density gradient (Fig. A3h) leading to a stronger AMOC

which increases SSTs and SSSs in the North Atlantic due to increased transport of heat and salt (orange markers Fig. 7a). Also here the salinity response is dominant for the density meaning the surface density increases. Because of the higher surface density, more freshwater forcing is necessary to block the isopycnals from outcropping which explains why $S_1$ shifts towards larger values of freshwater forcing (Fig. 4b). For positive biases freshening of the Atlantic basin causes the opposite response to the one described above (red markers Fig. 7).

The situation is more complicated when biases are introduced in both basins. Negative biases in the Atlantic Ocean cause an increase in salinity and density in the North Atlantic increasing the AMOC strength. Since the AMOC is exporting freshwater out of the Atlantic Ocean ($F_{ov,S} < 0$), the increased AMOC causes a decrease in salinity in the South Atlantic. This is effect

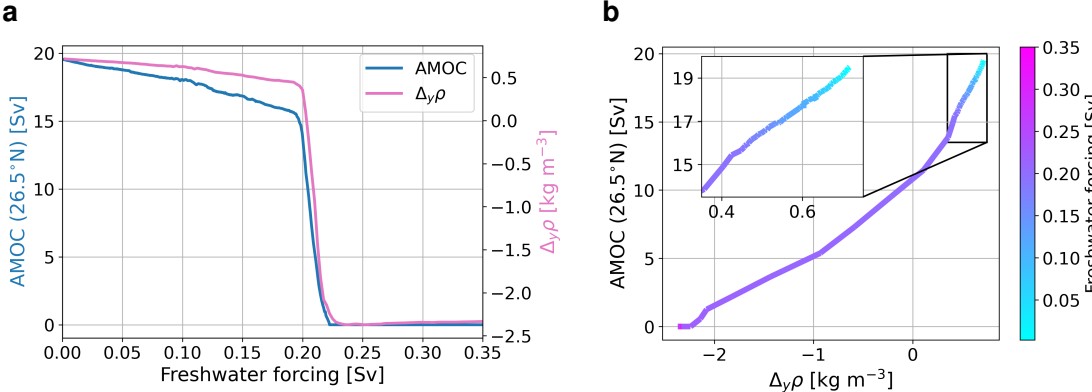

**Figure 6.** (a) AMOC strength at $26.5°N$ in Sv (blue; left y-axis) and the meridional density difference $\Delta_y\rho$ in kg m$^{-3}$ (pink; right y-axis) versus freshwater forcing $F_H$ in Sv (x-axis) for case BASE. (b) AMOC-$\Delta_y\rho$ relation. The inset is a zoom in before the AMOC collapse.

dominates over the effects of a more saline Indian Ocean. What makes these simulations different, is that the surface density in the North Atlantic is relatively independent of the strength and sign of the biases (except for the IA(+3.00) experiment) which

can be seen in Fig. 7 as the olive and green markers are located on the same isopycnal as BASE (black marker). This means that the changes in salinity and temperature caused by a different AMOC strength compensate each other for the surface density in the IA experiments. Since the surface density is similar, the amount of freshwater forcing necessary to block the isopycnals from outcropping is also similar explaining why $S_1$ hardly moves in the IA experiments (Fig. 4c).

## 4   Summary and discussion

This study complements previous research (Jackson et al., 2023b) in showing the importance of freshwater biases for the AMOC. Specifically, we have looked at the effect of freshwater biases, as identified in CMIP6 models, on AMOC hysteresis behavior in CLIMBER-X. We find that biases in the Indian and Atlantic Ocean can shift the collapse tipping point in hysteresis experiments. Positive biases in the Indian Ocean (i.e. freshening) lead to a shift of the tipping point towards larger freshwater forcing. Positive biases in the Atlantic Ocean have an opposite effect, i.e. the tipping point shifts towards smaller freshwater

forcing. When biases are introduced in both basins, the collapse tipping point does not show a shift.

In this study we have focused on buoyancy forcing in the North Atlantic to collapse the AMOC. However, Southern Ocean processes, such as eddies and upwelling, play an important role in shaping and driving the Global Overturning Circulation and the AMOC (Kuhlbrodt et al., 2007). In higher resolution models, these Southern Ocean processes can prevent the AMOC from fully collapsing by sustaining a very weak and shallow AMOC. In Baker et al. (2025), they suggest that the AMOC can only

fully collapse if the Southern Ocean upwelling is compensated for by an emerging Pacific Meridional Overturning Circulation (PMOC), which in the CMIP6 models they analysed was not the case. In our study, however, the upwelling is compensated for by changes in the Southern Ocean overturning circulation and a strong PMOC allowing the AMOC to fully collapse. In

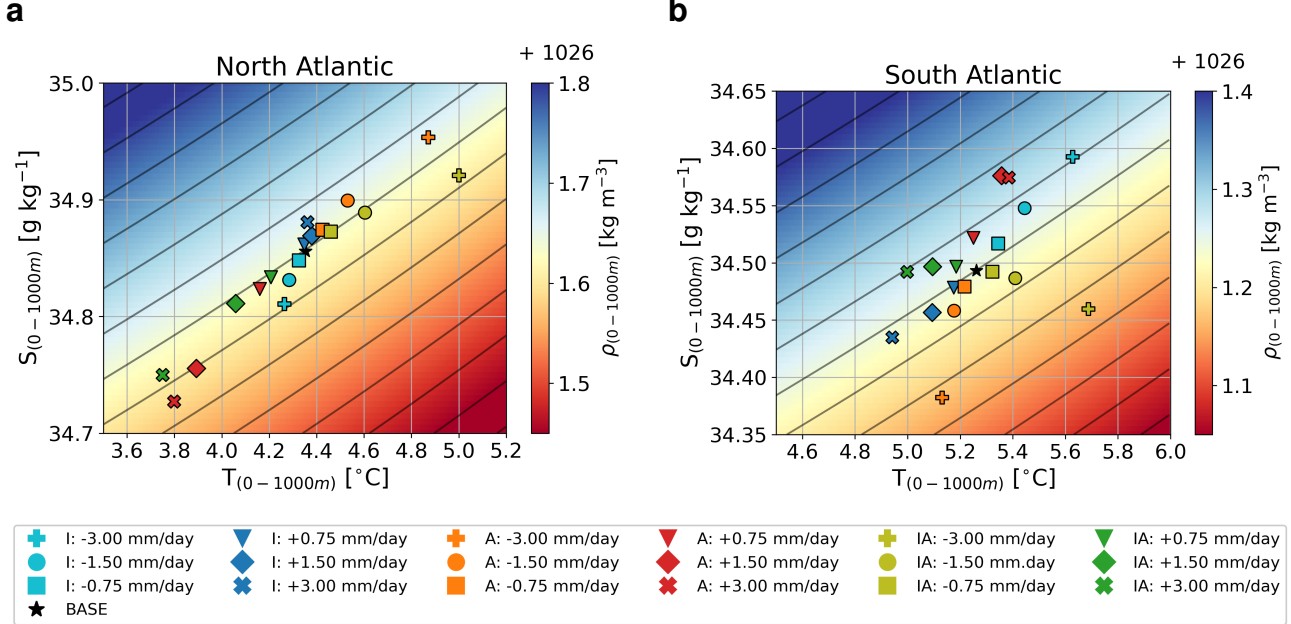

**Figure 7.** Sea Surface Temperature in °C (x-axis) versus Sea Surface Salinity in g kg$^{-1}$ (y-axis). Coloring in the background represents surface density in kg m$^{-3}$ and the contour lines are isopycnals. The data points represent the variables at the start of the hysteresis experiments at a freshwater forcing of 0 Sv. (a) For the North Atlantic box. (b) For the South Atlantic box.

a strongly eddying ocean-only model, using a similar simulation protocol as used in this study, van Westen et al. (2025) find that the AMOC does not fully collapse which might be attributed to eddies in the Southern Ocean. However, the differences in climate impact between a fully collapsed AMOC and a very weak AMOC are small.

The width of the hysteresis of the baseline case (BASE) compares well to other models (Rahmstorf et al., 2005) and previous studies with CLIMBER-X (Willeit et al., 2022; Willeit and Ganopolski, 2024). The exact hysteresis width is dependent on (among others) the hosing rate. The rate chosen in this study is the same as in Rahmstorf et al. (2005). In Willeit and Ganopolski (2024) also a slower hosing rate is used which results in a narrower hysteresis width with the collapse tipping point at lower freshwater forcing. However, we do not expect that a slower hosing rate would lead to different conclusions for our study as a similar shift due to a specific bias would occur.

Although such biases have been found in CMIP6 models, it is unfortunately not yet possible to make any convincing statements on how these biases would affect AMOC stability in these models. One reason is that the background states of the CMIP6 models and that of CLIMBER-X are very different and hence also the effects of biases are expected to be different. Another source of uncertainty is the fact that the CLIMBER-X simulations were performed in a stable pre-industrial climate state, whereas the biases determined in the CMIP6 models are from short, transient simulations under historical forcing. There is no way to circumvent this, since we only have observations in the historical period and the hysteresis experiments need to be

done in quasi-equilibrium. How this influences the conclusions of this study we cannot say. The biases are also determined over a relatively short period. This means that if (strong) multidecadal variability is present in the freshwater fluxes, the assessed

bias strength in the CMIP6 models is not a good representation of the actual model biases. Finally, besides freshwater biases, there are also biases in sea ice and surface air temperatures that may influence AMOC stability, and are not considered in this study.

Because of the coarse resolution and the simple atmospheric model in CLIMBER-X, atmospheric feedbacks might not be as important in our simulations compared to CMIP6 models. The importance of changes in atmospheric heat and moisture

transport on hysteresis simulations was, for example, already shown in Jackson et al. (2016) in a different EMIC with a higher ocean resolution than CLIMBER-X. As mentioned earlier, Southern Ocean upwelling is important for shaping the AMOC as well, and CMIP6 models show a change in the Southern Ocean westerlies following an AMOC weakening which changes the Ekman dynamics (Madan et al., 2023; Baker et al., 2025). A weakening AMOC might also decrease the wind stress curl over the North Atlantic subpolar gyre (Madan et al., 2023), which could act as positive feedback during an AMOC collapse

by decreasing deep convection. These atmospheric feedbacks can play a different role in the CMIP6 models compared to CLIMBER-X and can therefore lead to a different response to freshwater biases in CMIP6 models compared to CLIMBER-X.

On the positive side, we presented a clear physical mechanism in Section 3.3, with an important role of the salt-advection feedback, on how the biases change the AMOC hysteresis properties in CLIMBER-X. These physical processes are also expected to be present in the CMIP6 type models. However, in CMIP6 models other processes, e.g. the atmospheric feedbacks

mentioned earlier, may become more important. To assess how important these other feedbacks are, hysteresis experiments with CMIP6 type models following a similar protocol as in this study should be performed. Although this is computationally challenging, we hope that this paper will stimulate simulations where the effects of biases are systematically addressed in these models.

*Code and data availability.* CMIP6 data can be downloaded from the Earth System Grid Federation (ESGF). All model data and scripts

necessary for the results presented in this study can be found at Boot (2025).

*Author contributions.* AAB and HAD conceptualized the study. AAB acquired the results. Both authors contributed to writing the manuscript.

*Competing interests.* The authors declare that they have no conflict of interest.

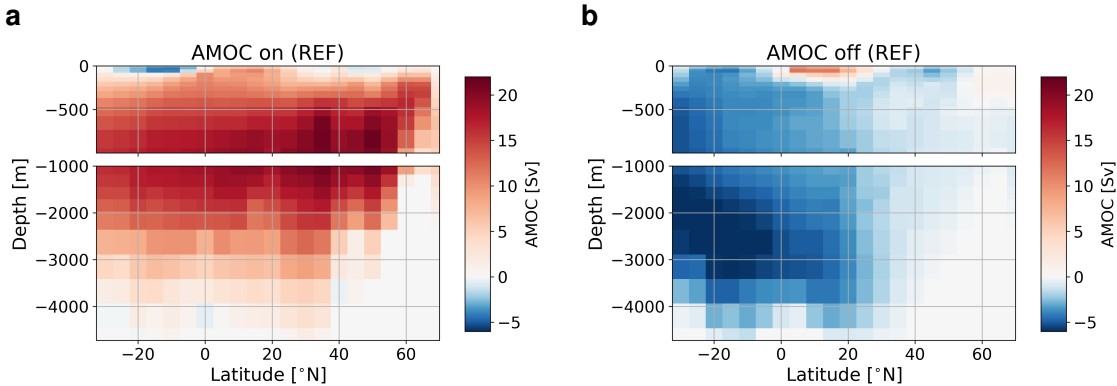

**Figure A1.** The AMOC in Sv in latitude (x-axis) - depth (y-axis) space for case BASE in (a) an on-state at a freshwater forcing of 0 Sv, and in (b) an off-state at a freshwater forcing of 0.35 Sv.

*Financial support.* This research has been supported by the European Research Council through the ERC-AdG project TAOC (PI: Dijkstra, project 101055096). The work was also partially supported by the ClimTip project, which has received funding from the European Union's Horizon Europe research and innovation programme under grant agreement No. 101137601.


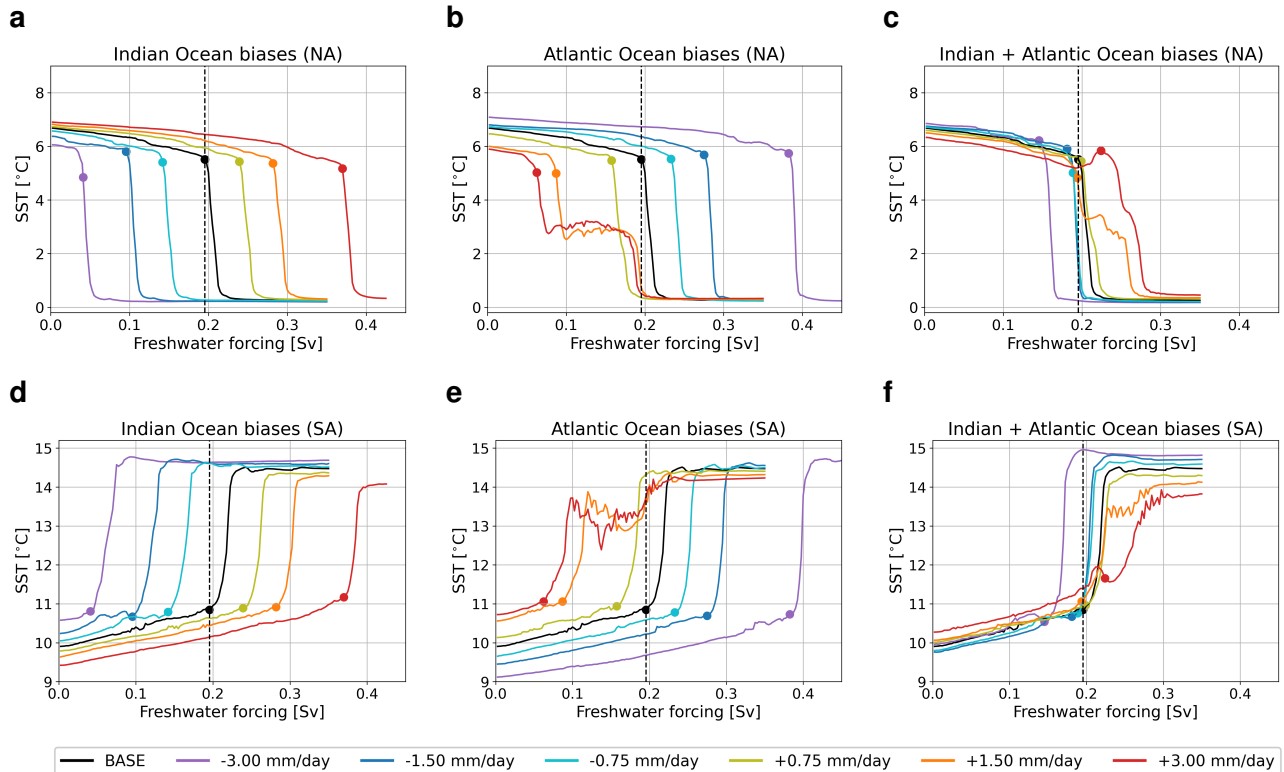

**Figure A2.** Sea Surface Temperature (SST) in °C in the North Atlantic (50°N - 70°N; 70°W - 25°E; a - c) and in the South Atlantic (35°S - 55°S; 55°W - 20°E; d - f). (a), (d) for simulations with biases in the Indian Ocean. (b), (e) for simulations with biases in the Atlantic Ocean. (c), (f) for simulations with biases in both basins. The markers represent the tipping points, and the black dashed line represents the tipping point of BASE.

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

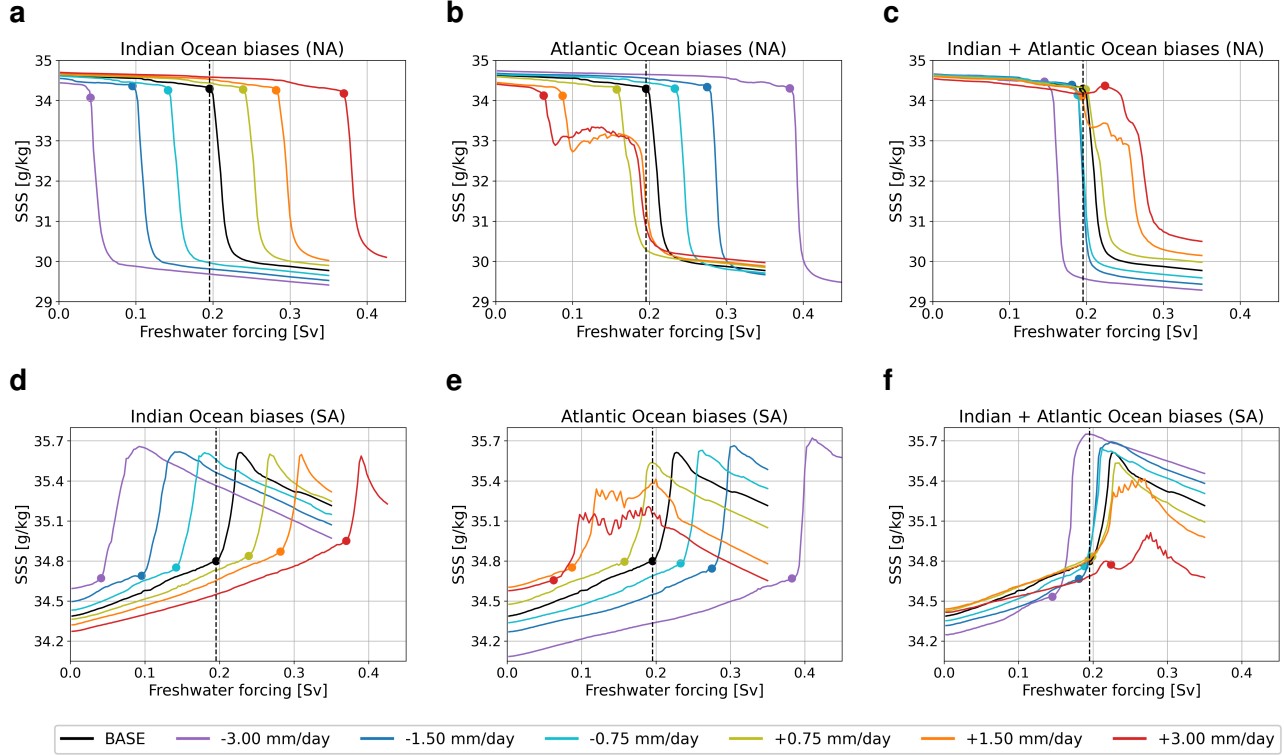

**Figure A3.** As Fig. A2 but for Sea Surface Salinity in g kg$^{-1}$.

Bentsen, M., Oliviè, D. J. L., Seland, Ø., Toniazzo, T., Gjermundsen, A., Graff, L. S., Debernard, J. B., Gupta, A. K., He, Y., Kirkevag, A., Schwinger, J., Tjiputra, J., Aas, K. S., Bethke, I., Fan, Y., Griesfeller, J., Grini, A., Guo, C., Ilicak, M., Karset, I. H. H., Landgren, O. A., Liakka, J., Moseid, K. O., Nummelin, A., Spensberger, C., Tang, H., Zhang, Z., Heinze, C., Iversen, T., and Schulz, M.: NCC NorESM2-MM model output prepared for CMIP6 CMIP historical, https://doi.org/10.22033/ESGF/CMIP6.8040, 2019.

Boot, A., von der Heydt, A. S., and Dijkstra, H. A.: Effect of Plankton Composition Shifts in the North Atlantic on Atmospheric pCO2, Geophysical Research Letters, 50, e2022GL100 230, https://doi.org/https://doi.org/10.1029/2022GL100230, 2023.

Boot, A. A.: ESD-fw-bias, https://doi.org/10.5281/zenodo.14887681, 2025.

Boot, A. A., von der Heydt, A. S., and Dijkstra, H. A.: Response of atmospheric pCO$_2$ to a strong AMOC weakening under low and high emission scenarios, Climate Dynamics, https://doi.org/10.1007/s00382-024-07295-y, 2024.

Boot, A. A., Steenbeek, J. G., Coll, M., von der Heydt, A. S., and Dijkstra, H. A.: Global marine ecosystem response to a strong AMOC weakening under low and high future emission scenarios, Earth's Future, https://doi.org/10.1029/2024EF004741, 2025.

Boucher, O., Denvil, S., Levavasseur, G., Cozic, A., Caubel, A., Foujols, M.-A., Meurdesoif, Y., Cadule, P., Devilliers, M., Ghattas, J., Lebas, N., Lurton, T., Mellul, L., Musat, I., Mignot, J., and Cheruy, F.: IPSL IPSL-CM6A-LR model output prepared for CMIP6 CMIP historical, https://doi.org/10.22033/ESGF/CMIP6.5195, 2018.

Bouillon, S., Ángel Morales Maqueda, M., Legat, V., and Fichefet, T.: An elastic–viscous–plastic sea ice model formulated on Arakawa B and C grids, Ocean Modelling, 27, 174–184, https://doi.org/https://doi.org/10.1016/j.ocemod.2009.01.004, 2009.

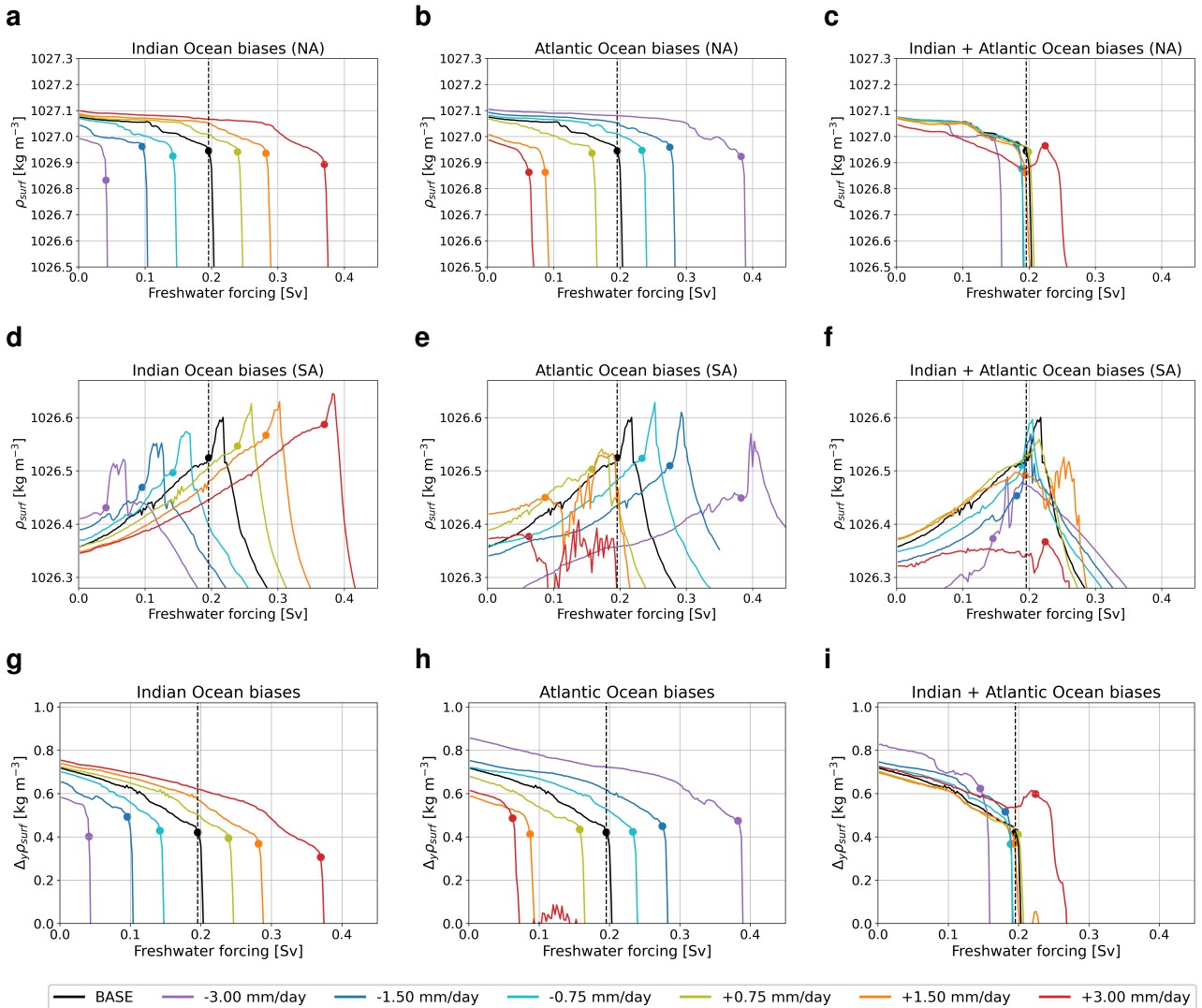

**Figure A4.** (a) - (f) as in Fig. A2 but for surface density in kg m$^{-3}$. (g) - (i) represent the density difference between the northern box and southern box ($\rho_N$ - $\rho_S$) in kg m$^{-3}$ for the Indian Ocean biases (g), the Atlantic Ocean biases (h), and the biases in both basins (i).

Broecker, W. S., Peteet, D. M., and Rind, D.: Does the ocean–atmosphere system have more than one stable mode of operation?, Nature, 315, 21–26, https://doi.org/10.1038/315021a0, 1985.

320 Bryden, H. L., King, B. A., and McCarthy, G. D.: South Atlantic overturning circulation at 24°S, Journal of Marine Research, 69, 38 – 55, 2011.

Caesar, L., Rahmstorf, S., Robinson, A., Feulner, G., and Saba, V.: Observed fingerprint of a weakening Atlantic Ocean overturning circulation, Nature, 556, 191–196, 2018.

**Table A1.** CMIP6 model list.

| Number | Name | Reference |
|---|---|---|
| 1. | ACCESS-CM2 | Dix et al. (2019) |
| 2. | ACCESS-ESM1-5 | Ziehn et al. (2019) |
| 3. | BCC-CSM2-MR | Wu et al. (2018) |
| 4. | CanESM5 | Swart et al. (2019b) |
| 5. | CanESM5-CanOE | Swart et al. (2019a) |
| 6. | CAS-ESM2-0 | Chai (2020) |
| 7. | CESM2 | Danabasoglu (2019a) |
| 8. | CESM2-FV2 | Danabasoglu (2019c) |
| 9. | CESM2-WACCM | Danabasoglu (2019b) |
| 10. | CMCC-CM2-SR5 | Lovato and Peano (2020) |
| 11. | CMCC-ESM2 | Lovato et al. (2021) |
| 12. | CNRM-CM6-1 | Voldoire (2018) |
| 13. | CNRM-CM6-1-HR | Voldoire (2019) |
| 14. | CNRM-ESM2-1 | Seferian (2018) |
| 15. | EC-Earth3-CC | Consortium (2021) |
| 16. | EC-Earth3-Veg-LR | Consortium (2020) |
| 17. | FGOALS-f3-L | Yu (2019) |
| 18. | FGOALS-g3 | Li (2019) |
| 19. | FIO-ESM-2-0 | Song et al. (2019) |
| 20. | GISS-E2-1-G | NASA/GISS (2018) |
| 21. | HadGEM3-GC31-LL | Ridley et al. (2019a) |
| 22. | HadGEM3-GC31-MM | Ridley et al. (2019b) |
| 23. | IPSL-CM6A-LR | Boucher et al. (2018) |
| 24. | MCM-UA-1-0 | Stouffer (2019) |
| 25. | MIROC-ES2L | Hajima et al. (2019) |
| 26. | MIROC6 | Tatebe and Watanabe (2018) |
| 27. | MPI-ESM1-2-HR | Jungclaus et al. (2019) |
| 28. | MRI-ESM2-0 | Yukimoto et al. (2019) |
| 29. | NESM3 | Cao and Wang (2019) |
| 30. | NorESM2-MM | Bentsen et al. (2019) |
| 31. | TaiESM1 | Lee and Liang (2020) |
| 32. | UKESM1-0-LL | Tang et al. (2019) |

Caesar, L., McCarthy, G. D., Thornalley, D. J. R., Cahill, N., and Rahmstorf, S.: Current Atlantic Meridional Overturning Circulation weakest
325    in last millennium, Nat. Geosci., 14, 118–120, 2021.

Cao, J. and Wang, B.: NUIST NESMv3 model output prepared for CMIP6 CMIP historical, https://doi.org/10.22033/ESGF/CMIP6.8769, 2019.

Chai, Z.: CAS CAS-ESM1.0 model output prepared for CMIP6 CMIP historical, https://doi.org/10.22033/ESGF/CMIP6.3353, 2020.

Consortium, E.-E.: EC-Earth-Consortium EC-Earth3-Veg-LR model output prepared for CMIP6 CMIP historical, https://doi.org/10.22033/ESGF/CMIP6.4707, 2020.

Consortium, E.-E.: EC-Earth-Consortium EC-Earth-3-CC model output prepared for CMIP6 CMIP historical, https://doi.org/10.22033/ESGF/CMIP6.4702, 2021.

Danabasoglu, G.: NCAR CESM2 model output prepared for CMIP6 CMIP historical, https://doi.org/10.22033/ESGF/CMIP6.7627, 2019a.

Danabasoglu, G.: NCAR CESM2-WACCM model output prepared for CMIP6 CMIP historical, https://doi.org/10.22033/ESGF/CMIP6.10071, 2019b.

Danabasoglu, G.: NCAR CESM2-FV2 model output prepared for CMIP6 CMIP historical, https://doi.org/10.22033/ESGF/CMIP6.11297, 2019c.

Danabasoglu, G., Yeager, S. G., Bailey, D., Behrens, E., Bentsen, M., Bi, D., Biastoch, A., Böning, C., Bozec, A., Canuto, V. M., Cassou, C., Chassignet, E., Coward, A. C., Danilov, S., Diansky, N., Drange, H., Farneti, R., Fernandez, E., Fogli, P. G., Forget, G., Fujii, Y., Griffies, S. M., Gusev, A., Heimbach, P., Howard, A., Jung, T., Kelley, M., Large, W. G., Leboissetier, A., Lu, J., Madec, G., Marsland, S. J., Masina, S., Navarra, A., George Nurser, A., Pirani, A., y Mélia, D. S., Samuels, B. L., Scheinert, M., Sidorenko, D., Treguier, A.-M., Tsujino, H., Uotila, P., Valcke, S., Voldoire, A., and Wang, Q.: North Atlantic simulations in Coordinated Ocean-ice Reference Experiments phase II (CORE-II). Part I: Mean states, Ocean Modelling, 73, 76–107, https://doi.org/https://doi.org/10.1016/j.ocemod.2013.10.005, 2014.

Dekker, M. M., von der Heydt, A. S., and Dijkstra, H. A.: Cascading transitions in the climate system, [dataset]. Earth System Dynamics, 9, 1243–1260, https://doi.org/10.5194/esd-9-1243-2018, 2018.

Dijkstra, H. A.: Characterization of the multiple equilibria regime in a global ocean model, Tellus A, 59, 695–705, https://doi.org/https://doi.org/10.1111/j.1600-0870.2007.00267.x, 2007.

Dijkstra, H. A. and van Westen, R. M.: The Effect of Indian Ocean Surface Freshwater Flux Biases On the Multi-Stable Regime of the AMOC, Tellus A: Dynamic Meteorology and Oceanography, https://doi.org/10.16993/tellusa.3246, 2024.

Dima, M. and Lohmann, G.: Evidence for Two Distinct Modes of Large-Scale Ocean Circulation Changes over the Last Century, Journal of Climate, 23, 5 – 16, https://doi.org/10.1175/2009JCLI2867.1, 2010.

Ditlevsen, P. and Ditlevsen, S.: Warning of a forthcoming collapse of the Atlantic meridional overturning circulation, Nature Communications, 14, 4254, 2023.

Dix, M., Bi, D., Dobrohotoff, P., Fiedler, R., Harman, I., Law, R., Mackallah, C., Marsland, S., O'Farrell, S., Rashid, H., Srbinovsky, J., Sullivan, A., Trenham, C., Vohralik, P., Watterson, I., Williams, G., Woodhouse, M., Bodman, R., Dias, F. B., Domingues, C. M., Hannah, N., Heerdegen, A., Savita, A., Wales, S., Allen, C., Druken, K., Evans, B., Richards, C., Ridzwan, S. M., Roberts, D., Smillie, J., Snow, K., Ward, M., and Yang, R.: CSIRO-ARCCSS ACCESS-CM2 model output prepared for CMIP6 CMIP historical, https://doi.org/10.22033/ESGF/CMIP6.4271, 2019.

Drijfhout, S. S., Weber, S. L., and van der Swaluw, E.: The stability of the MOC as diagnosed from model projections for pre-industrial, present and future climates, Clim. Dyn., 37, 1575–1586, 2011.

Edwards and Shepherd: Bifurcations of the thermohaline circulation in a simplified three-dimensional model of the world ocean and the effects of inter-basin connectivity, Clim. Dyn., 19, 31–42, 2002.

Edwards, N. R. and Marsh, R.: Uncertainties due to transport-parameter sensitivity in an efficient 3-D ocean-climate model, Clim. Dyn., 24, 415–433, 2005.

Edwards, N. R., Willmott, A. J., and Killworth, P. D.: On the Role of Topography and Wind Stress on the Stability of the Thermohaline Circulation, Journal of Physical Oceanography, 28, 756 – 778, https://doi.org/10.1175/1520-0485(1998)028<0756:OTROTA>2.0.CO;2, 1998.

Garzoli, S. L., Baringer, M. O., Dong, S., Perez, R. C., and Yao, Q.: South Atlantic meridional fluxes, Deep Sea Research Part I: Oceanographic Research Papers, 71, 21–32, https://doi.org/https://doi.org/10.1016/j.dsr.2012.09.003, 2013.

Haines, K., Ferreira, D., and Mignac, D.: Variability and Feedbacks in the Atlantic Freshwater Budget of CMIP5 Models With Reference to Atlantic Meridional Overturning Circulation Stability, Frontiers in Marine Science, 9, https://doi.org/10.3389/fmars.2022.830821, 2022.

Hajima, T., Abe, M., Arakawa, O., Suzuki, T., Komuro, Y., Ogura, T., Ogochi, K., Watanabe, M., Yamamoto, A., Tatebe, H., Noguchi, M. A., Ohgaito, R., Ito, A., Yamazaki, D., Ito, A., Takata, K., Watanabe, S., Kawamiya, M., and Tachiiri, K.: MIROC MIROC-ES2L model output prepared for CMIP6 CMIP historical, https://doi.org/10.22033/ESGF/CMIP6.5602, 2019.

Hawkins, E., Smith, R. S., Allison, L. C., Gregory, J. M., Woollings, T. J., Pohlmann, H., and de Cuevas, B.: Bistability of the Atlantic overturning circulation in a global climate model and links to ocean freshwater transport, Geophysical Research Letters, 38, https://doi.org/https://doi.org/10.1029/2011GL047208, 2011.

Heuzé, C.: Antarctic Bottom Water and North Atlantic Deep Water in CMIP6 models, Ocean Science, 17, 59–90, https://doi.org/10.5194/os-17-59-2021, 2021.

Hunke, E. C. and Dukowicz, J. K.: An Elastic–Viscous–Plastic Model for Sea Ice Dynamics, Journal of Physical Oceanography, 27, 1849 – 1867, https://doi.org/10.1175/1520-0485(1997)027<1849:AEVPMF>2.0.CO;2, 1997.

Intergovernmental Panel on Climate Change (IPCC): Climate change 2021 – the physical science basis, Cambridge University Press, Cambridge, England, 2023.

Jackson, L. C. and Petit, T.: North Atlantic overturning and water mass transformation in CMIP6 models, Clim. Dyn., 2022.

Jackson, L. C., Kahana, R., Graham, T., Ringer, M. A., Woollings, T., Mecking, J. V., and Wood, R. A.: Global and European climate impacts of a slowdown of the AMOC in a high resolution GCM, Clim. Dyn., 45, 3299–3316, 2015.

Jackson, L. C., Smith, R. S., and Wood, R. A.: Ocean and atmosphere feedbacks affecting AMOC hysteresis in a GCM, Climate Dynamics, 49, 173–191, https://doi.org/10.1007/s00382-016-3336-8, 2016.

Jackson, L. C., Alastrué de Asenjo, E., Bellomo, K., Danabasoglu, G., Haak, H., Hu, A., Jungclaus, J., Lee, W., Meccia, V. L., Saenko, O.,
Shao, A., and Swingedouw, D.: Understanding AMOC stability: the North Atlantic Hosing Model Intercomparison Project, Geoscientific Model Development, 16, 1975–1995, https://doi.org/10.5194/gmd-16-1975-2023, 2023a.

Jackson, L. C., Hewitt, H. T., Bruciaferri, D., Calvert, D., Graham, T., Guiavarc'h, C., Menary, M. B., New, A. L., Roberts, M., and Storkey, D.: Challenges simulating the AMOC in climate models, Philosophical Transactions of the Royal Society A: Mathematical, Physical and Engineering Sciences, 381, 20220187, https://doi.org/10.1098/rsta.2022.0187, 2023b.

Jungclaus, J., Bittner, M., Wieners, K.-H., Wachsmann, F., Schupfner, M., Legutke, S., Giorgetta, M., Reick, C., Gayler, V., Haak, H., de Vrese, P., Raddatz, T., Esch, M., Mauritsen, T., von Storch, J.-S., Behrens, J., Brovkin, V., Claussen, M., Crueger, T., Fast, I., Fiedler, S., Hagemann, S., Hohenegger, C., Jahns, T., Kloster, S., Kinne, S., Lasslop, G., Kornblueh, L., Marotzke, J., Matei, D., Meraner, K., Mikolajewicz, U., Modali, K., Müller, W., Nabel, J., Notz, D., Peters-von Gehlen, K., Pincus, R., Pohlmann, H., Pongratz, J., Rast, S., Schmidt, H., Schnur, R., Schulzweida, U., Six, K., Stevens, B., Voigt, A., and Roeckner, E.: MPI-M MPI-ESM1.2-HR model output
prepared for CMIP6 CMIP historical, https://doi.org/10.22033/ESGF/CMIP6.6594, 2019.

Killick, R., Fearnhead, P., and Eckley, I.: Optimal Detection of Changepoints With a Linear Computational Cost, Journal of the American Statistical Association, 107, 1590–1598, https://doi.org/10.1080/01621459.2012.737745, 2012.

Kuhlbrodt, T., Griesel, A., Montoya, M., Levermann, A., Hofmann, M., and Rahmstorf, S.: On the driving processes of the Atlantic meridional overturning circulation, Reviews of Geophysics, 45, https://doi.org/https://doi.org/10.1029/2004RG000166, 2007.

Latif, M., Sun, J., Visbeck, M., and Hadi Bordbar, M.: Natural variability has dominated Atlantic Meridional Overturning Circulation since 1900, Nat. Clim. Chang., 12, 455–460, 2022.

Lee, W.-L. and Liang, H.-C.: AS-RCEC TaiESM1.0 model output prepared for CMIP6 CMIP historical, https://doi.org/10.22033/ESGF/CMIP6.9755, 2020.

Lenton, T. M., Held, H., Kriegler, E., Hall, J. W., Lucht, W., Rahmstorf, S., and Schellnhuber, H. J.: Tipping elements in the Earth's climate
system, Proceedings of the National Academy of Sciences, 105, 1786–1793, https://doi.org/10.1073/pnas.0705414105, 2008.

Levermann, A., Griesel, A., Hofmann, M., Montoya, M., and Rahmstorf, S.: Dynamic sea level changes following changes in the thermohaline circulation, Clim. Dyn., 24, 347–354, 2005.

Li, L.: CAS FGOALS-g3 model output prepared for CMIP6 CMIP historical, https://doi.org/10.22033/ESGF/CMIP6.3356, 2019.

Liu, W., Liu, Z., and Brady, E. C.: Why is the AMOC Monostable in Coupled General Circulation Models?, Journal of Climate, 27, 2427 –
2443, https://doi.org/10.1175/JCLI-D-13-00264.1, 2014.

Liu, W., Xie, S.-P., Liu, Z., and Zhu, J.: Overlooked possibility of a collapsed Atlantic Meridional Overturning Circulation in warming climate, Science Advances, 3, e1601 666, https://doi.org/10.1126/sciadv.1601666, 2017.

Lobelle, D., Beaulieu, C., Livina, V., Sévellec, F., and Frajka-Williams, E.: Detectability of an AMOC Decline in Current and Projected Climate Changes, Geophysical Research Letters, 47, e2020GL089 974, https://doi.org/https://doi.org/10.1029/2020GL089974,
e2020GL089974 10.1029/2020GL089974, 2020.

Lovato, T. and Peano, D.: CMCC CMCC-CM2-SR5 model output prepared for CMIP6 CMIP historical, https://doi.org/10.22033/ESGF/CMIP6.3825, 2020.

Lovato, T., Peano, D., and Butenschön, M.: CMCC CMCC-ESM2 model output prepared for CMIP6 CMIP historical, https://doi.org/10.22033/ESGF/CMIP6.13195, 2021.

Lynch-Stieglitz, J.: The Atlantic Meridional Overturning Circulation and Abrupt Climate Change, Annual Review of Marine Science, 9, 83–104, https://doi.org/10.1146/annurev-marine-010816-060415, 2017.

Madan, G., Gjermundsen, A., Iversen, S. C., and LaCasce, J. H.: The weakening AMOC under extreme climate change, Climate Dynamics, 62, 1291–1309, https://doi.org/10.1007/s00382-023-06957-7, 2023.

McCarthy, G. D. and Caesar, L.: Can we trust projections of AMOC weakening based on climate models that cannot reproduce
the past?, Philosophical Transactions of the Royal Society A: Mathematical, Physical and Engineering Sciences, 381, 20220 193, https://doi.org/10.1098/rsta.2022.0193, 2023.

Mecking, J., Drijfhout, S., Jackson, L., and Andrews, M.: The effect of model bias on Atlantic freshwater transport and implications for AMOC bi-stability, Tellus A: Dynamic Meteorology and Oceanography, https://doi.org/10.1080/16000870.2017.1299910, 2017.

Michel, S. L. L., Swingedouw, D., Ortega, P., Gastineau, G., Mignot, J., McCarthy, G., and Khodri, M.: Early warning signal for a tipping
point suggested by a millennial Atlantic Multidecadal Variability reconstruction, Nat. Commun., 13, 5176, 2022.

Müller, S. A., Joos, F., Edwards, N. R., and Stocker, T. F.: Water Mass Distribution and Ventilation Time Scales in a Cost-Efficient, Three-Dimensional Ocean Model, Journal of Climate, 19, 5479 – 5499, https://doi.org/10.1175/JCLI3911.1, 2006.

NASA/GISS: NASA-GISS GISS-E2.1G model output prepared for CMIP6 CMIP historical, https://doi.org/10.22033/ESGF/CMIP6.7127, 2018.

Orihuela-Pinto, B., England, M. H., and Taschetto, A. S.: Interbasin and interhemispheric impacts of a collapsed Atlantic Overturning Circulation, Nature Climate Change, 12, 558–565, https://doi.org/10.1038/s41558-022-01380-y, 2022.

Rahmstorf, S.: On the freshwater forcing and transport of the Atlantic thermohaline circulation, Climate Dynamics, 12, 799–811, https://doi.org/10.1007/s003820050144, 1996.

Rahmstorf, S., Crucifix, M., Ganopolski, A., Goosse, H., Kamenkovich, I., Knutti, R., Lohmann, G., Marsh, R., Mysak, L. A.,
Wang, Z., and Weaver, A. J.: Thermohaline circulation hysteresis: A model intercomparison, Geophysical Research Letters, 32, https://doi.org/https://doi.org/10.1029/2005GL023655, 2005.

Rahmstorf, S., Box, J. E., Feulner, G., Mann, M. E., Robinson, A., Rutherford, S., and Schaffernicht, E. J.: Exceptional twentieth-century slowdown in Atlantic Ocean overturning circulation, Nat. Clim. Chang., 5, 475–480, 2015.

Ridley, J., Menary, M., Kuhlbrodt, T., Andrews, M., and Andrews, T.: MOHC HadGEM3-GC31-LL model output prepared for CMIP6 CMIP
historical, https://doi.org/10.22033/ESGF/CMIP6.6109, 2019a.

Ridley, J., Menary, M., Kuhlbrodt, T., Andrews, M., and Andrews, T.: MOHC HadGEM3-GC31-MM model output prepared for CMIP6 CMIP historical, https://doi.org/10.22033/ESGF/CMIP6.6112, 2019b.

Romanou, A., Rind, D., Jonas, J., Miller, R., Kelley, M., Russell, G., Orbe, C., Nazarenko, L., Latto, R., and Schmidt, G. A.: Stochastic Bifurcation of the North Atlantic Circulation under a Midrange Future Climate Scenario with the NASA-GISS ModelE, Journal of Climate,
36, 6141 – 6161, https://doi.org/10.1175/JCLI-D-22-0536.1, 2023.

Rossby, T., Palter, J., and Donohue, K.: What can hydrography between the New England slope, Bermuda and Africa tell us about the strength of the AMOC over the last 90 years?, Geophys. Res. Lett., 49, 2022.

Schmittner, A.: Decline of the marine ecosystem caused by a reduction in the Atlantic overturning circulation, Nature, 434, 628–633, https://doi.org/10.1038/nature03476, 2005.

Seferian, R.: CNRM-CERFACS CNRM-ESM2-1 model output prepared for CMIP6 CMIP historical, https://doi.org/10.22033/ESGF/CMIP6.4068, 2018.

Sinet, S., von der Heydt, A. S., and Dijkstra, H. A.: AMOC Stabilization Under the Interaction With Tipping Polar Ice Sheets, Geophysical Research Letters, 50, e2022GL100 305, https://doi.org/https://doi.org/10.1029/2022GL100305, 2023.

Smeed, D. A., Josey, S. A., Beaulieu, C., Johns, W. E., Moat, B. I., Frajka-Williams, E., Rayner, D., Meinen, C. S., Baringer, M. O., Bryden,
H. L., and McCarthy, G. D.: The North Atlantic Ocean Is in a State of Reduced Overturning, Geophysical Research Letters, 45, 1527–1533, https://doi.org/https://doi.org/10.1002/2017GL076350, 2018.

Song, Z., Qiao, F., Bao, Y., Shu, Q., Song, Y., and Yang, X.: FIO-QLNM FIO-ESM2.0 model output prepared for CMIP6 CMIP historical, https://doi.org/10.22033/ESGF/CMIP6.9199, 2019.

Stommel, H.: Thermohaline Convection with Two Stable Regimes of Flow, Tellus, 13, 224–230,
https://doi.org/https://doi.org/10.1111/j.2153-3490.1961.tb00079.x, 1961.

Stouffer, R.: UA MCM-UA-1-0 model output prepared for CMIP6 CMIP historical, https://doi.org/10.22033/ESGF/CMIP6.8888, 2019.

Stouffer, R. J., Yin, J., Gregory, J. M., Dixon, K. W., Spelman, M. J., Hurlin, W., Weaver, A. J., Eby, M., Flato, G. M., Hasumi, H., Hu, A., Jungclaus, J. H., Kamenkovich, I. V., Levermann, A., Montoya, M., Murakami, S., Nawrath, S., Oka, A., Peltier, W. R., Robitaille, D. Y., Sokolov, A., Vettoretti, G., and Weber, S. L.: Investigating the Causes of the Response of the Thermohaline Circulation to Past and Future
Climate Changes, Journal of Climate, 19, 1365 – 1387, https://doi.org/10.1175/JCLI3689.1, 2006.

Swart, N. C., Cole, J. N. S., Kharin, V. V., Lazare, M., Scinocca, J. F., Gillett, N. P., Anstey, J., Arora, V., Christian, J. R., Jiao, Y., Lee, W. G., Majaess, F., Saenko, O. A., Seiler, C., Seinen, C., Shao, A., Solheim, L., von Salzen, K., Yang, D., Winter, B., and Sigmond, M.: CCCma CanESM5-CanOE model output prepared for CMIP6 CMIP historical, https://doi.org/10.22033/ESGF/CMIP6.10260, 2019a.

Swart, N. C., Cole, J. N. S., Kharin, V. V., Lazare, M., Scinocca, J. F., Gillett, N. P., Anstey, J., Arora, V., Christian, J. R., Jiao, Y., Lee, W. G., Majaess, F., Saenko, O. A., Seiler, C., Seinen, C., Shao, A., Solheim, L., von Salzen, K., Yang, D., Winter, B., and Sigmond, M.: CCCma CanESM5 model output prepared for CMIP6 CMIP historical, https://doi.org/10.22033/ESGF/CMIP6.3610, 2019b.

Tang, Y., Rumbold, S., Ellis, R., Kelley, D., Mulcahy, J., Sellar, A., Walton, J., and Jones, C.: MOHC UKESM1.0-LL model output prepared for CMIP6 CMIP historical, https://doi.org/10.22033/ESGF/CMIP6.6113, 2019.

Tatebe, H. and Watanabe, M.: MIROC MIROC6 model output prepared for CMIP6 CMIP historical, https://doi.org/10.22033/ESGF/CMIP6.5603, 2018.

Terhaar, J., Vogt, L., and Foukal, N. P.: Atlantic overturning inferred from air-sea heat fluxes indicates no decline since the 1960s, Nat. Commun., 16, 222, 2025.

Tian, B. and Dong, X.: The Double-ITCZ Bias in CMIP3, CMIP5, and CMIP6 Models Based on Annual Mean Precipitation, Geophysical Research Letters, 47, e2020GL087 232, https://doi.org/https://doi.org/10.1029/2020GL087232, e2020GL087232 2020GL087232, 2020.

van Westen, R. M. and Dijkstra, H. A.: Asymmetry of AMOC Hysteresis in a State-Of-The-Art Global Climate Model, Geophysical Research Letters, 50, e2023GL106 088, https://doi.org/https://doi.org/10.1029/2023GL106088, e2023GL106088 2023GL106088, 2023.

van Westen, R. M. and Dijkstra, H. A.: Persistent climate model biases in the Atlantic Ocean's freshwater transport, Ocean Science, 20, 549–567, https://doi.org/10.5194/os-20-549-2024, 2024.

van Westen, R. M., Kliphuis, M., and Dijkstra, H. A.: Physics-based early warning signal shows that AMOC is on tipping course, Science Advances, 10, eadk1189, https://doi.org/10.1126/sciadv.adk1189, 2024.

van Westen, R. M., Kliphuis, M., and Dijkstra, H. A.: Collapse of the Atlantic Meridional Overturning Circulation in a Strongly Eddying Ocean-Only Model, Geophysical Research Letters, 52, e2024GL114 532, https://doi.org/https://doi.org/10.1029/2024GL114532, e2024GL114532 2024GL114532, 2025.

Vanderborght, E., van Westen, R. M., and Dijkstra, H. A.: Feedback Processes causing an AMOC Collapse in the Community Earth System Model, https://arxiv.org/abs/2410.03236, 2024.

Vellinga, M. and Wood, R. A.: Impacts of thermohaline circulation shutdown in the twenty-first century, Climatic Change, 91, 43–63, https://doi.org/10.1007/s10584-006-9146-y, 2008.

Voldoire, A.: CMIP6 simulations of the CNRM-CERFACS based on CNRM-CM6-1 model for CMIP experiment historical, https://doi.org/10.22033/ESGF/CMIP6.4066, 2018.

Voldoire, A.: CNRM-CERFACS CNRM-CM6-1-HR model output prepared for CMIP6 CMIP historical, https://doi.org/10.22033/ESGF/CMIP6.4067, 2019.

Weijer, W., Cheng, W., Drijfhout, S. S., Fedorov, A. V., Hu, A., Jackson, L. C., Liu, W., McDonagh, E. L., Mecking, J. V., and Zhang, J.: Stability of the Atlantic Meridional Overturning Circulation: A Review and Synthesis, Journal of Geophysical Research: Oceans, 124, 5336–5375, https://doi.org/https://doi.org/10.1029/2019JC015083, 2019.

Weijer, W., Cheng, W., Garuba, O. A., Hu, A., and Nadiga, B. T.: CMIP6 Models Predict Significant 21st Century Decline of the Atlantic Meridional Overturning Circulation, Geophysical Research Letters, 47, e2019GL086 075, https://doi.org/https://doi.org/10.1029/2019GL086075, 2020.

Willeit, M. and Ganopolski, A.: PALADYN v1.0, a comprehensive land surface–vegetation–carbon cycle model of intermediate complexity, Geoscientific Model Development, 9, 3817–3857, https://doi.org/10.5194/gmd-9-3817-2016, 2016.

Willeit, M. and Ganopolski, A.: Generalized stability landscape of the Atlantic meridional overturning circulation, [dataset]. Earth System Dynamics, 15, 1417–1434, https://doi.org/10.5194/esd-15-1417-2024, 2024.

Willeit, M., Ganopolski, A., Robinson, A., and Edwards, N. R.: The Earth system model CLIMBER-X v1.0 – Part 1: Climate model description and validation, Geoscientific Model Development, 15, 5905–5948, https://doi.org/10.5194/gmd-15-5905-2022, 2022.

Wolfe, C. L. and Cessi, P.: Multiple Regimes and Low-Frequency Variability in the Quasi-Adiabatic Overturning Circulation, Journal of
Physical Oceanography, 45, 1690 – 1708, https://doi.org/10.1175/JPO-D-14-0095.1, 2015.

Worthington, E. L., Moat, B. I., Smeed, D. A., Mecking, J. V., Marsh, R., and McCarthy, G. D.: A 30-year reconstruction of the Atlantic meridional overturning circulation shows no decline, Ocean Science, 17, 285–299, https://doi.org/10.5194/os-17-285-2021, 2021.

Wu, T., Chu, M., Dong, M., Fang, Y., Jie, W., Li, J., Li, W., Liu, Q., Shi, X., Xin, X., Yan, J., Zhang, F., Zhang, J., Zhang, L., and Zhang, Y.: BCC BCC-CSM2MR model output prepared for CMIP6 CMIP historical, https://doi.org/10.22033/ESGF/CMIP6.2948, 2018.

Wunderling, N., von der Heydt, A. S., Aksenov, Y., Barker, S., Bastiaansen, R., Brovkin, V., Brunetti, M., Couplet, V., Kleinen, T., Lear, C. H., Lohmann, J., Roman-Cuesta, R. M., Sinet, S., Swingedouw, D., Winkelmann, R., Anand, P., Barichivich, J., Bathiany, S., Baudena, M., Bruun, J. T., Chiessi, C. M., Coxall, H. K., Docquier, D., Donges, J. F., Falkena, S. K. J., Klose, A. K., Obura, D., Rocha, J., Rynders, S., Steinert, N. J., and Willeit, M.: Climate tipping point interactions and cascades: a review, [dataset]. Earth System Dynamics, 15, 41–74, https://doi.org/10.5194/esd-15-41-2024, 2024.

Yin, J., Griffies, S. M., and Stouffer, R. J.: Spatial Variability of Sea Level Rise in Twenty-First Century Projections, Journal of Climate, 23, 4585 – 4607, https://doi.org/10.1175/2010JCLI3533.1, 2010.

Yu, Y.: CAS FGOALS-f3-L model output prepared for CMIP6 CMIP historical, https://doi.org/10.22033/ESGF/CMIP6.3355, 2019.

Yukimoto, S., Koshiro, T., Kawai, H., Oshima, N., Yoshida, K., Urakawa, S., Tsujino, H., Deushi, M., Tanaka, T., Hosaka, M., Yoshimura, H., Shindo, E., Mizuta, R., Ishii, M., Obata, A., and Adachi, Y.: MRI MRI-ESM2.0 model output prepared for CMIP6 CMIP historical,
https://doi.org/10.22033/ESGF/CMIP6.6842, 2019.

Zickfeld, K., Eby, M., and Weaver, A. J.: Carbon-cycle feedbacks of changes in the Atlantic meridional overturning circulation under future atmospheric CO2, Global Biogeochemical Cycles, 22, https://doi.org/https://doi.org/10.1029/2007GB003118, 2008.

Ziehn, T., Chamberlain, M., Lenton, A., Law, R., Bodman, R., Dix, M., Wang, Y., Dobrohotoff, P., Srbinovsky, J., Stevens, L., Vohralik, P., Mackallah, C., Sullivan, A., O'Farrell, S., and Druken, K.: CSIRO ACCESS-ESM1.5 model output prepared for CMIP6 CMIP historical,
https://doi.org/10.22033/ESGF/CMIP6.4272, 2019.