# Peer review of "Physics of AMOC multistable regime shifts due to freshwater biases in an EMIC"

_EGUsphere, 2025_

## Author Comment (AC1)

**MS-No.: egusphere-2025-758**

**Title: Physics of AMOC Multistable Regime Shifts due to Freshwater Biases in an EMIC**

**Authors:** Amber A. Boot and Henk A. Dijkstra

**Point-by-point reply to reviewer #1**

May 7, 2025

We thank Susanne Ditlevsen for their careful reading and their generous review.

**General comments:**

*It is of great importance to understand the risks of tipping of the Atlantic Meridional Overturning Circulation (AMOC). This is often done using climate models, however, these are known to have biases, in particular, freshwater biases in the Indian and the Atlantic Ocean, which might affect the model evaluations of AMOC stability. It is therefore of great interest to quantify how such biases might affect model outputs. This is the goal of the paper. The paper conducts larger simulation studies of CLIMBER-X, an Earth System Model of intermediate complexity to study the effect of biases in surface freshwater flux on AMOC tipping behavior. Several scenarios of biases are introduced in the Indian and Atlantic Ocean, as well as the reference level with no bias. Then they perform hysteresis experiments on all scenarios, where the surface freshwater forcing is slowly ramped up in the North Atlantic until the AMOC collapses; subsequently, the forcing is reversed until the AMOC recovers again.*

*The paper shows that the AMOC stability is hugely affected by freshwater biases. This is an important result, and underpins the importance of being careful when drawing quantitative conclusions from climate models regarding tipping elements, in particular the AMOC.*

*The paper is very well written, the methods well chosen and executed and statements, conclusions, methods and goals clearly detailed. Figures are of*

*high quality.*

*Congratulations with a really nice work.*
**Technical corrections:**

*It is confusing with the notation REF for the reference model. It looks like there is an error with a reference. This is not important, just a suggestion to change the notation.*

**Author's reply:**
We agree that the current notation can lead to confusion.

**Changes in manuscript:**
We will rename the reference case to baseline case and use the abbreviation BASE instead of REF.

---

## Author Comment (AC2)

**MS-No.: egusphere-2025-758**

**Title: Physics of AMOC Multistable Regime Shifts due to Freshwater Biases in an EMIC**

**Authors:** Amber A. Boot and Henk A. Dijkstra

**Point-by-point reply to reviewer #2**

May 7, 2025

We thank R. Marsh for their careful reading and their useful comments on the manuscript.

**General comments:**

*The authors have undertaken a focussed study of AMOC hysteresis for a plausible (CMIP-informed) range of biases in surface freshwater forcing over the Indian and Atlantic oceans. Using an Earth System Model of Intermediate Complexity, CLIMBER-X, a substantial impact on AMOC stability is evident, a result that should be of interest to those engaged in a wide range of AMOC monitoring, modelling and related research.*

*Following the pioneering study of Stommel (1961), the interplay of freshwater forcing and transport was highlighted more recently by Rahmstorf (1996), which along with emerging paleo evidence (Broecker 2010, and references therein) attracted wider interest to the issue of AMOC stability. This sub-field has since developed incrementally over the last 30 years, and this manuscript is a useful contribution to our understanding of model dependence of AMOC hysteresis.*

*It appears from Fig. 1a that the 'REF' configuration of CLIMBER-X has a bistable AMOC, in that there are two stable states (on and off) at Freshwater Forcing = 0 Sv. This is noteworthy, as are monotable or bistable AMOCs evident in subsequent 18 hysteresis experiments. This aspect of AMOC stability is central to the issue of hysteresis, S1, S2 and H, worthy of comment in results and discussion.*

*The authors are appropriately cautious in discussion, not least due to the limitations of CLIMBER-X, which likely lacks key feedbacks. In particular,*

*the imposed freshwater fluxes (over Atlantic and Indian oceans) are held fixed throughout the experiments. This is highly artificial, as one might expect tele-connected changes to E-P across the global ocean, as part of the coupled response to a collapsing (or recovering) AMOC. Also implicit in this study is the longstanding assumption that the AMOC is buoyancy forced from the north, while others have long argued that the AMOC is mechanically forced from the south (reviewed by Kuhlbrodt et al. 2007). Given the here-acknowledged importance of changes in the SA, and the NA-SA density difference, are feedbacks involving the Southern Ocean - specifically wind-driven and eddy-mediated dense water upwelling - worthy of note?*

**Author's reply:**
There is no assumption in our paper that the AMOC is buoyancy forced from the north. However, what is important here is that buoyancy forcing in the North Atlantic can collapse the AMOC. This happens because the shared outcropping isopycnals between the Southern Ocean and North Atlantic that enable adiabatic transport in the interior are disconnected due to the reduced surface density in the North Atlantic. The importance for this can be seen in Figure 5a where the meridional density gradient changes sign when the AMOC collapses. Furthermore, as the AMOC collapses in our simulations, there is still wind-driven upwelling. However, this is compensated for by adjustments in the Global Overturning Circulation including changes in Southern Ocean overturning, and a strengthening of the PMOC.

This does not mean that in different models, and especially models with higher resolution, the same happens. In an eddying ocean-only using a similar simulation strategy (van Westen et al., 2025) the AMOC collapse to a weak state where there is still a small overturning cell in the Atlantic Ocean. Southern Ocean eddies might play an important role in maintaining a weak AMOC in this case. Also in Baker et al. (2025) it is suggested that Southern Ocean upwelling can prevent an AMOC collapse if it is not compensated for by an emerging PMOC.

**Changes in manuscript:**
We will include a discussion on Southern Ocean processes in Section 4.

*The manuscript is succinctly written, with well-crafted figures that convey a rich level of information. I close with the following specific comments:*

1. *Introduction: References to the earlier/earliest studies of AMOC hysteresis and stability (see above) would be appropriate, in the opening part of the Introduction.*

   **Author's reply:**
   We agree and we thank the reviewer for their suggestions.

   **Changes in manuscript:**
   Earlier studies of AMOC hysteresis will be added to the introduction.

2. *Sect. 2.2: In the hysteresis experiment, freshwater forcing in the Atlantic, in the zone 20-50N, is increased/decreased at 0.05 Sv/yr; later in the discussion, this is briefly justified and discussed, but it would be appropriate to justify in Sect. 2.2, also the zone (notably south of convection sites).*

   **Author's reply:**
   The justification for the hosing region (i.e. 20°N - 50°N) is twofold. (1) As far as we know this region is used by most other studies that perform quasi-equilibrium hosing experiments. Most notably are the studies of Rahmstorf et al. (2005) and, more recently, van Westen and Dijkstra (2023). (2) hosing more northward would mean we would be hosing over the convection sites as the reviewer points out. This would mean that blocking the outcropping isopycnals in the North Atlantic by direct freshwater forcing might (partly) mask the salt-advection feedback.

   **Changes in manuscript:**
   A brief justification will be added to Section 2.2.

3. *Sect. 2.2 / Fig. 2: Where is Fig. 2 referenced in the main text? This would naturally be at lines 111-115.*

   **Author's reply:**
   Figure 2 is indeed not referenced in the text yet. We agree with the

suggestion.

**Changes in manuscript:**
We will refer to Figure 2 in the suggested paragraph.

4. *Sect. 3.3, lines 221-222: Analysis of density compensation of changes in salinity and temperature in the IA experiments needs some elaboration; I inferred that the Atlantic bias primarily affects the NA while the Indian bias affects SA, in opposite senses – is this correct?*

   **Author's reply:**
   A negative bias in the Atlantic Ocean mostly influences the North Atlantic box, causing an increase in salinity and density there. This increases the AMOC strength. Since the AMOC is exporting freshwater out of the Atlantic Ocean ($F_{ov,S} < 0$; Fig. 3), the stronger AMOC causes a decrease in salinity in the South Atlantic. Negative biases in the Indian Ocean act to increase the salinity in the South Atlantic. However, the increased AMOC strength ensures that the salinity in the South Atlantic decreases and a new balance is found.

   **Changes in manuscript:**
   We will expand the discussion in the text.

5. *Summary and discussion, lines 251-252: Regarding 'other processes, e.g., atmospheric feedbacks', there is scope to expand on this to discuss the effects of changing atmospheric heat and moisture transports, wind stress curl (NA subpolar gyre) and Ekman dynamics (Southern Ocean), on the AMOC (collapsing or recovering).*

   **Author's reply:**
   We agree.

   **Changes in manuscript:**
   A more elaborate discussion will be included.

**References**

- Rahmstorf, S., Crucifix, M., Ganopolski, A., Goosse, H., Kamenkovich, I., Knutti, R., Lohmann, G., Marsh, R., Mysak, L. A., Wang, Z., and Weaver, A. J.: Thermohaline circulation hysteresis: A model intercomparison, Geophysical Research Letters, 32, https://doi.org/https://doi.org/10.1029/2005GL02 2005.

- van Westen, R. M. and Dijkstra, H. A.: Asymmetry of AMOC Hysteresis in a State-Of-The-Art Global Climate Model, Geophysical Research Letters, 50, e2023GL106 088, https://doi.org/https://doi.org/10.1029/2023GL106088, e2023GL106088 2023GL106088, 2023.